# Overt and occult hepatitis B virus infection detected among chronic kidney disease patients on haemodialysis at a Tertiary Hospital in Ghana

Bartholomew Dzudzor[1]*, Kingsley Kwame Nsowah[1], Seth Agyemang[2], Sandro Vento[3], Vincent Amarh[1], Vincent Boima[4], Kenneth Tachi[4]*

1 Department of Medical Biochemistry, University of Ghana Medical School, Korle-Bu, Ghana, 2 Yemaachi Biotechnology, Accra, Ghana, 3 Faculty of Medicine, University of Puthisastra, Phnom Penh, Cambodia, 4 Department of Medicine and Therapeutics, University of Ghana Medical School, Korle-Bu, Ghana

* bdzudzor@ug.edu.gh (BD); ktachi@ug.edu.gh (KT)

## Abstract

Hepatitis B virus (HBV) infection is endemic in Ghana and chronic kidney disease patients on haemodialysis are a high-risk group for HBV infection. We determined the prevalence of overt and occult HBV infection among haemodialysis patients at the Korle Bu Teaching Hospital in Ghana. 104 consenting End Stage Renal Disease patients on long-term haemodialysis were recruited for the study and their socio-demographic, clinical and laboratory information were obtained using structured questionnaire. All the participants were tested for the hepatitis B surface antigen (HBsAg). The HBsAg-negative participants were re-tested for hepatitis B surface antibody (HBsAb), hepatitis B core antibody (HBcAb) and HBV DNA using chemiluminescence and Roche COBAS Ampli-Prep/TaqMan analyser and real-time polymerase chain reaction. Eight (7.7%) of the total participants were positive for HBsAg. Among the 96 HBsAg-negative participants, 12.5% (12) were HBcAb-positive, 7.3% (7) had detectable HBV DNA (mean = 98.7±53.5 IU/mL) and 40.6% (39) were positive for HBsAb. Five out of the 7 HBV DNA-positive participants were males and only one participant was negative for HBcAb. Seventy-three out of the 96 HBsAg-negative participants were vaccinated and 37 of these vaccinated individuals had significant HBsAb titres (mean = 423.21± 380.72 IU/mL). Our data demonstrated that the prevalence of overt and occult HBV infection among the haemodialysis (HD) patients was 7.7% and 7.3%, respectively, and only 50.7% of those who showed proof of vaccination were protected from HBV infection.

## Introduction

Globally, 2 billion people are estimated to have past or present infection with HBV, and 240 million people are chronic carriers of HBsAg, with the highest endemicity in sub-Saharan Africa [1]. Seventy percent (70%) of hepatitis-associated mortality is attributed to HBV

**Data Availability Statement:** All relevant data are within the paper and its Supporting Information files.

**Funding:** -KT and BD -Partially funded by: University of Ghana Research Fund (UGRF/10/ILG-081/2016-2017) awarded to KT (main investigator) and BD (co-investigator). -No. The funders had no role in study design, data collection and analysis, decision to publish, or preparation of the manuscript.

**Competing interests:** The authors have declared that no competing interest exist.

infection [2]. Furthermore, approximately 800,000 annual deaths worldwide are due to HBV-associated complications [3].

The estimated prevalence of HBV in Ghanaian population is 12.3%, thereby placing Ghana among the highly endemic countries [4]. In areas of such high HBV endemicity, most of the HBV infection is typically acquired in childhood, either horizontally from other children, or vertically from the mother [5]. Among those vulnerable to horizontal transmission of HBV are patients and staff in dialysis units. This occurrence is due to frequent blood transfusions, possible contamination of dialysis equipment and other sources of environmental contamination [6]. Acute HBV infections in dialysis patients tend to be mild and asymptomatic, and therefore likely to be unrecognized, with increased risk of progression to chronicity [4,7]. Defective innate and adaptive immunity in chronic kidney disease (CKD) patients, result in about 60% of acute infection progressing to chronic carrier state compared to 5% in the general population [8]. The increased probability of chronicity in haemodialysis (HD) patients increases the pool of HBV in HD units, and leads to increased risk of chronic liver disease, premature death from cirrhosis or liver cancer, and nosocomial transmission within the HD units [9].

Unfortunately, HBV infection has not received the needed public health attention compared to HIV and, more recently, SARS-CoV2. In Ghana, there has been a steady rise in the number of persons with CKD undergoing chronic dialysis because of improved accessibility. However, there have been no studies from Ghana on the prevalence of HBV in dialysis patients. Due to resource constraints, strict adherence to standard HD infection control practices is challenging, and will consequently cause further increase in the risk of nosocomial transmissions within dialysis units in Ghana. Additionally, many end-stage renal disease patients prefer multiple transfusions to the relatively costly erythropoiesis stimulating treatments such as erythropoietin, thereby exacerbating their exposure to the risk of HBV infection.

HBV vaccination has been utilized to effectively control infection, resulting in 70% reduction in the risk of HBV infection in vaccinated patients within dialysis units compared to unvaccinated patients [10]. Poor response to vaccination [10], and faster rate of decay of anti-HBs titres remain a major challenge for protection in HD patients [8]. Routine checking of antibody titres every 6-12months is recommended with re-vaccination, if required [11]. However, adherence to this recommendation is challenging in resource-limiting countries, where patients remain at a high risk of being infected. Currently in Ghana, patients are tested only when they start haemodialysis, contrary to international guidelines. Moreover, those who test negative for HBsAg are encouraged to be vaccinated at their own expense and there is no insistence on follow-up testing to confirm the development of immunity. The lack of guidelines implies efforts at preventing nosocomial HBV transmission are not coordinated, and an infection of one HD patient can lead to an uncontrolled HBV outbreak within dialysis units in Ghana.

Another potential source of HBV transmission in HD units are carriers of occult HBV infection (OBI). OBI refers to HBsAg-negative patients with HBV DNA in liver and/or serum, with or without HBcAb [12]. Regardless of the fact that HBV transmissibility from patients with OBI remains a subject of intense research and debate, this condition can increase the risk of liver cancer by eight-fold [13]. The clinical outcome of HBV transmission primarily depends on the recipient's immune status and the number of HBV DNA copies present in the blood. The presence of donor HBsAb reduces the risk of HBV infection by approximately five-fold [14]. The geographical distribution of occult HBV infection is similar to overt HBV infection [15]. Studies on HD patients have provided varied results that mirrored the level of HBV endemicity in different areas. In Iran, 50% prevalence of HBV DNA was detected among HD patients identified as HBcAb positive/HBsAb negative [16]. In contrast, HBV DNA was not detected in 34 HBcAb positive haemodialysis patients in Brazil [17] and a prevalence of 2.2%

was reported in UK haemodialysis patients [15]. Data on OBI among CKD patients undergoing HD is not available in Ghana. Therefore, this study is focused on determining overt HBV infection, immune-protection status and occult hepatitis B virus infection among CKD patients on HD at Korle Bu Teaching Hospital. This was achieved by testing for HBsAg, HBcAb, HBsAb and HBV DNA to evaluate HBV infection, previous exposure to HBV, acquired immunity through vaccination and OBI respectively.

## Methods

### Study design

This was a hospital-based cross-sectional study. One hundred and four (104) consenting patients, clinically diagnosed with end-stage renal disease, on haemodialysis at the Renal Unit of the Korle-Bu Teaching Hospital were initially enrolled in the study between September 17, and December 21, 2017. Demographic data (age and gender) were obtained from all the 104 patients using a structured questionnaire. The structured questionnaire and the patients' folders were further used to obtain additional information including the cause of the CKD, duration on haemodialysis and status of HBV vaccination and blood transfusion. Five millilitres (5 ml) of venous blood were also obtained from these patients through venepuncture during a single dialysis session and the serum was separated into microtubes for analysis. The serum was initially used to test for hepatitis B surface antigen (HBsAg) to determine the prevalence of overt HBV infection among these HD patients. The sera of the patients were also used to test for the levels of HBcAb and HBsAb to identify patients with a past exposure to the HBV and non-immune patients (HBsAb < 10 IU/mL), respectively. The levels of HBV DNA in the serum of the HBsAg-negative patients were quantified to identify occult infections.

### Laboratory procedure

The Rapid Diagnostic Test strip Alere Determine (Alere Medical Co. Ltd., Tokyo, Japan), that has a sensitivity and specificity of 97.2% and 98.5% with an analytical sensitivity of 0.1 IU/mL and a lower limit of detection of 0.05%, was first used to screen for the HBsAg status of the 104 haemodialysis patients [18,19]. The manufacturer's protocol was followed except 60μl of serum were used instead of 50μl. The test kits, serum samples, and buffer were all equilibrated to room temperature prior to the testing. A second confirmatory test was performed for the HBsAg-positive samples using Electrochemiluminescence assay reagents (Elecsys HBsAg II) from Roche Diagnostics (Hamburg, Germany) with a sensitivity of 100% and specificity of 99.88%. The Elecsys HBsAg II Electrochemiluminescence assay has a cut-off index < 0.9 for non-reactive samples and an analytical sensitivity ≤ 0.04 IU/mL. Determination of antibody responses to hepatitis B core antigen (anti-HBc) and hepatitis B surface antigen (anti-HBs) were also performed using Elecsys anti HBs II assay, which has a limit of detection of 0.02 IU/mL. Confirmatory HBV DNA was performed using reagents from Roche COBAS (CAP-CTM Roche Molecular Systems, Inc. Brachburg, NJ) platform, according to manufacturer's instructions.

### HBV DNA quantification

The COBAS AmpliPrep/TaqMan HBV test (CAP-CTM Roche Molecular Systems, Inc. Brachburg, NJ), which is a fully automated system consisting of two integrated platforms: the COBAS AmpliPrep for automated nucleic acid extraction from plasma specimens and the COBAS TaqMan 48 (a real-time PCR assay based on Taqman technology) was used to quantify serum HBV DNA. The COBAS AmpliPrep-COBAS Taqman HBV test is an automated real-time PCR test based on a dual-labelled hybridization probe targeting the pre-core and core regions of HBV

DNA. This automated DNA extraction is based on the affinity of DNA for silica gel covered magnetic beads. An internal quantitation standard (QS) was added to each sample during the processing step. After extraction of HBV DNA with the COBAS AmpliPrep instrument, the COBAS TaqMan 48 analyzer was used to perform the real-time PCR test via a multiplex Taq-Man assay. The sensitivity of the COBAS AmpliPrep/TaqMan HBV test is 10 IU/ml [20].

The sequences of the primers used for the amplification and detection of the S-core region of HBV are as follows:

Forward primer: 5'-CTCCCCGTCTGTGCCTTCTCATC-3' nucleotide (1545–1567) and the reverse primer: 5'-GGCGTTCACGGTGGTCTCCATGC-3' nucleotide (1606–1628). Two targets were amplified: HBV DNA and the internal QS yielding an amplicon size of 83bp.

## Ethical consideration

The principles of the Declaration of Helsinki and its appendices, as well as national legislation, were followed during the conduct of the study. The College of Health Sciences' Ethical and Protocol Review Committee of the University of Ghana approved this study with reference number (CHS-Et/M. 11-P 4.9/2016-2017), and all patients enrolled into the study endorsed a written consent form. Confidentiality was protected by assigning unique code to patients' demographic data and ensuring restricted access to the original database.

## Statistical analysis

Data from the study were analysed in SPSS version 21. Demographic characteristics were presented as means, standard deviations and percentages. Chi-square test and Odds Ratio were used to determine association of the clinical parameters and p-value $< 0.05$ was considered statistically significant. Data were analysed by categorizing patients into immune (HBsAb$\geq$10.0 IU/mL) and non-immune (HBsAb$<$10.0 IU/mL), immunity from vaccination and immunity from past infection, and occult infection and no occult infection.

## Results

A total of 104 HD patients comprising of 55.8% males and 44.2% females were initially recruited for the study. The prevalence of HBsAg-positive patients was 7.7% (8/104), and consisted of 6 males and 2 females. Of the HBsAg-positive patients, 4 (50%) had HD duration of less than one year, 7 had a record of hypertension being the cause of CKD and none of them showed any evidence of vaccination. Majority of the HBsAg-positive patients were 40 years and older (Table 1). The 96 HBsAg-negative patients comprised of 52 males and 44 females, and the age range was 21 to 75 years (mean age = 45.2 ± 13.9 years). Hypertension was the main cause of CKD, accounting for 68.8% (66/96), followed by glomerulonephritis (18.8%). The mean duration of haemodialysis for the HBsAg-negative patients was 2.3 ± 2.0 years (Table 2). Eighty-five (88.5%) of patients had received blood transfusion and 76% (73/96) of the patients showed proof of vaccination (Table 2).

Thirty-nine (40.6%) out of the 96 HBsAg-negative patients had baseline immunity against HBV (HBsAb > 10 IU/mL), of which 22 were males and 17 were females. The HBsAb titres ranged from 21 to 1311 IU/mL, with a mean of 423.21± 380.72 IU/mL. HBsAb titres were > 10UI/mL in only 50.7% (37/73) out of the 73 vaccinated patients. In addition, 8.7% (2/23) of the unvaccinated patients had HBsAb > 10 IU/mL through previous exposure to HBV infection, since these patients were also positive for HBcAb (Table 3). Among patients with prior vaccination (HBsAb+, HBcAb-), 35 (41.7%) had immunity against HBsAg compared to 4 (33.3%) of the patients with previous HBV infection (HBsAb+, HBcAb+). The proportion of HBsAb > 10 IU/mL was highest among patients on dialysis for a year or more (Table 3).

**Table 1. Demographic characteristics of the haemodialysis patients.**

| Variable | HbsAg status | | Total |
|---|---|---|---|
| | Negative N (%) | Positive N (%) | N |
| **Age group (years)** | | | |
| 20–29 | 18(90) | 2(10) | 20 |
| 30–39 | 14(100) | 0(0.0) | 14 |
| 40–49 | 30(90.9) | 3(9.1) | 33 |
| 50–59 | 15(93.8) | 1(6.2) | 16 |
| 60+ | 19(90.5) | 2(9.5) | 21 |
| Total | 96(92.3) | 8(7.7) | 104 |
| **Sex** | | | |
| Female | 44(95.7) | 2(4.3) | 46 |
| Male | 52(89.7) | 6(10.3) | 58 |
| **Total** | **96(92.3)** | **8(7.7)** | **104** |
| **Duration of HD for HBsAg+** | | | |
| | <1 | **4(50)** | |
| | >2 | **1(12.5)** | |
| | 3+ | **1(12.5)** | |
| | Not sure | **2(25)** | |
| **Cause of CKD for HBsAg+** | | | |
| | Hypertension | **7(87.5)** | |
| | Multiple myeloma | **1(12.5)** | |
| **Blood transfusion for HBsAg+** | | | |
| | No | **0** | |
| | Yes | **8(100)** | |
| **Vaccination for HBsAg+** | | | |
| | | No records found for all of them | |
| **HbsAg level in HBsAg+** | | | |
| **Mean±SD (IU/mL)** | | **6374.63±2121.14** | |
| **Range (IU/mL)** | | **1895–8379** | |

Of the 96 HBsAg-negative patients, 52 were males and 44 females. Eighty-five (88.5%) of the 96 HBsAg-negatives had been transfused with blood and 11 (11.5%) had not. Also 76% (73/96) showed proof of vaccination and 23 (24%) had not (Tables 2 and 3).

Only 39 (40.6%) had baseline immunity against HBV (HBsAb > 10 IU/mL), of which 22 were males and 17 females out of the 96 HD patients. The HBsAb titres ranged from 21 to 1311 IU/mL with a mean of 423.21± 380.72 IU/mL. Of the 73 vaccinated patients, HBsAb titres were > 10UI/mL in only 50.7% (37/73). In addition, 8.7% (2/23) of the unvaccinated also had HBsAb > 10 IU/mL through previous exposure to HBV infection, since these patients were also HBcAb-positive (Table 3). Among patients with prior vaccination (HBsAb+, HBcAb-), 35 (41.7%) had immunity against HBsAg compared to 4 (33.3%) of patients with previous HBV infection (HBsAb+, HBcAb+). The proportion of HBsAb > 10 IU/mL was highest among patients on dialysis for a year or more (Table 3).

The prevalence of previous HBV infection (HBcAb-positive) was 12.5% and consisted of 7 males and 5 females (Table 4). One-third of the HBsAg-negative patients with prior exposure to HBV (HBcAb-positive) also reacted to HBsAb, indicating immunity through natural infection. Moreover, ten out of the 12 HBcAb-positive patients had a history of blood transfusion (Table 4). Our data also demonstrated that HBcAb was a statistically significant predictor of occult hepatitis B infection among the CKD patients (Table 5).

**Table 2. Demographic and clinical characteristics of the HBsAg-negative participants.**

| Variable | N (%) |
|---|---|
| **Gender** | |
| Female | 44 (45.8%) |
| Male | 52 (54.2%) |
| **Age (Years) [Mean (SD)]** | **45.2 ± 13.9** |
| **Age groups (years)** | |
| 20–29 | 19 (19.8) |
| 30–39 | 13 (13.5) |
| 40–49 | 29 (30.2) |
| 50–59 | 16 (16.7) |
| 60+ | 19 (19.8) |
| **Duration of Hemodialysis (Years)** | |
| <1 | 25 (26.0) |
| 1–2 | 42 (43.8) |
| 3+ | 29 (30.2) |
| **Cause of CKD** | |
| Glomerulonephritis | 18 (18.8) |
| Hypertension | 66 (68.8) |
| Diabetes | 1 (1.0) |
| Lupus | 3 (3.1) |
| Other | 8 (8.3) |
| **Blood transfusion** | |
| No | 11 (11.5) |
| Yes | 85 (88.5) |
| **Vaccination** | |
| No | 23 (24.0) |
| Yes | 73 (76.0) |

Total N = 96.

Occult HBV infection was detected in 7 (7.3%) out of the 96 HBsAg-negative patients and they consisted of 5 males and 2 females, with a viral load ranging from 34 to 193 IU/mL and a mean of 98.7±53.5 IU/mL (Table 6). All the OBI patients had HBV viral load less than 200 IU/mL, which is indicative of a true OBI phenotype [12]. Furthermore, 6 out of the 7 OBI patients tested positive for HBcAb and 2 OBI patients were positive for both HBcAb and HBsAb (Table 6). The youngest OBI patient was negative for both HBcAb and HBsAb, therefore being a case of seronegative OBI (Table 6).

Only 3 OBI patients had been vaccinated and 6 patients also confirmed they had received blood transfusion (Table 7). Most of the OBI patients were sixty years (4/7) of age or older, had been on haemodialysis for up to two years or were diagnosed with hypertension (Table 7). Prior infection with HBV (HBcAb-positive patients) and HBV vaccination were significantly associated with the OBI phenotype in this study (Table 7).

## Discussion

In this study, we report the first baseline data on the prevalence of chronic overt and occult hepatitis B virus infection among chronic kidney disease patients undergoing haemodialysis in Ghana. The immune status of the patients against HBV infection was also evaluated among

**Table 3. HBsAb immunity status of the HBsAg-negative patients.**

| Variable | Immune status (HBsAb) | | | | Total |
|---|---|---|---|---|---|
| | Non-reactive | | Reactive | | |
| | Frequency | Percentage | Frequency | Percentage | |
| **Sex** | | | | | |
| Female | 27 | 61.4 | 17 | 38.6 | 44 |
| Male | 30 | 57.7 | 22 | 42.3 | 52 |
| **Age group** | | | | | |
| 21–29 | 14 | 73.7 | 5 | 26.3 | 19 |
| 30–39 | 7 | 53.8 | 6 | 46.2 | 13 |
| 40–49 | 17 | 58.6 | 12 | 41.4 | 29 |
| 50–59 | 11 | 68.8 | 5 | 31.3 | 16 |
| 60+ | 8 | 42.1 | 11 | 57.9 | 19 |
| **Duration on haemodialysis** | | | | | |
| <1 | 18 | 72 | 7 | 28 | 25 |
| 1–2 | 24 | 57.1 | 18 | 42.9 | 42 |
| 3+ | 15 | 51.7 | 14 | 48.3 | 29 |
| **Cause of CKD** | | | | | |
| Glomerulonephritis | 13 | 72.2 | 5 | 27.8 | 18 |
| Hypertension | 34 | 51.5 | 32 | 48.5 | 66 |
| Other | 10 | 83.3 | 2 | 16.7 | 12 |
| **Blood transfusion** | | | | | |
| No | 8 | 72.7 | 3 | 27.3 | 11 |
| Yes | 49 | 57.6 | 36 | 42.4 | 85 |
| **Vaccination** | | | | | |
| No | 21 | 91.3 | 2 | 8.7 | 23 |
| Yes | 36 | 49.3 | 37 | 50.7 | 73 |
| **Core antibody** | | | | | |
| Non-reactive | 49 | 58.3 | 35 | 41.7 | 84 |
| Reactive | 8 | 66.7 | 4 | 33.3 | 12 |
| **HBsAb levels** | | | | | |
| Mean±SD (IU/mL) | | | 423.21±380.72 | | |
| Range (IU/mL) | | | 21–1311 | | |

the study subjects. Determination of the prevalence of HBV infection among HD patients is vital for prevention of nosocomial infections in dialysis units.

Our data are in agreement with a previous study conducted in Punjab, Pakistan [21]. In the United States and Western Europe a prevalence of less than 7% was reported in a study involving approximately 900 HD patients from 308 dialysis facilities [22]. Other studies from the United States, Europe, Japan and Libya reported HBV infection prevalence of 2.4%, 4.1%, 2.2% and 2.6%, respectively [23]. In, Ghana, it was reported that HBV prevalence in the general population ranges from 3.5% to 13.3% [4]. Our observations from this study were contrary to our expectations because we found lower prevalence in a HD Unit as compared to the general population, as previously reported. Nonetheless, our findings are similar to a prior study which found low prevalence rate of HBV (1.2%) in HD subjects compared to 7.3% in the general population [24]. Research conducted in developing nations estimated that the carriers of HBsAg among HD patients range from 2% to 20% [25,26]. The relatively lower prevalence of HBV infection among the HD patients at the Korle Bu Teaching Hospital than the general

**Table 4.  HbcAb status of the HBsAg-negative patients.**

| Variable | HBcAb status | | Total | chi-square(p-value) |
|---|---|---|---|---|
|  | Non-reactive | Reactive |  |  |
|  | n = 84(87.5) | n = 12(12.5) | N = 96 |  |
| **Sex** |  |  |  | 0.10(0.757) |
| Female | 39(88.6) | 5(11.4) | 44 |  |
| Male | 45(86.5) | 7(13.5) | 52 |  |
| **Age group** |  |  |  | 2.50(0.645) |
| 21–29 | 18(94.7) | 1(5.3) | 19 |  |
| 30–39 | 12(92.3) | 1(7.7) | 13 |  |
| 40–49 | 25(86.2) | 4(13.8) | 29 |  |
| 50–59 | 14(87.5) | 2(12.5) | 16 |  |
| 60+ | 15(78.9) | 4(21.1) | 19 |  |
| **Duration** |  |  |  | 1.24(0.539) |
| <1 | 21(84.0) | 4(16.0) | 25 |  |
| 1–2 | 36(85.7) | 6(14.3) | 42 |  |
| 3+ | 27(93.1) | 2(6.9) | 29 |  |
| **Cause of CKD** |  |  |  | 0.48(0.785) |
| G.N | 15(83.3) | 3(16.7) | 18 |  |
| Hypertension | 58(87.9) | 8(12.1) | 66 |  |
| Other | 11(91.7) | 1(8.3) | 12 |  |
| **Blood transfusion** |  |  |  | 0.37(0.545) |
| No | 9(81.8) | 2(18.2) | 11 |  |
| Yes | 75(88.2) | 10(11.8) | 85 |  |
| **Vaccination** |  |  |  | 8.90(0.003) |
| No | 16(69.6) | 7(30.4) | 23 |  |
| Yes | 68(93.2) | 5(6.8) | 73 |  |
| **HBcAb level** |  |  |  |  |
| Mean ±SD (IU/mL) |  | 3450.67± 2794.41 |  |  |
| Range (IU/mL) |  | 312–8313 |  |  |

population in Ghana may perhaps be partly attributed to the awareness of the vulnerability of the patients, with consequent improved control practices at the Renal Unit.

Our data indicate that only 50.7% of HBV vaccinated patients were HBsAb- positive and therefore protected from HBV infection. Our findings agree with a study in China where a rapid decay of HBsAb was observed in the HD patients and more importantly, the decline of immunoprotection depended on duration of haemodialysis [27]. Our data also confirm the findings of a study conducted in Brazil, where it was observed that the vaccine coverage of hepatitis B in haemodialysis patients was high but the serological profile of previous immunization did not confer an effective immunization status in these individuals, as only approximately 30% of them had isolated positivity for HBsAb [28]. The discrepancy between vaccination reports and low frequency of isolated positivity for HBsAb probably reflects the decline in these antibodies over time [29]. In fact, immunocompromised HD patients may have deficient humoral and cell-mediated immune responses, which leads to lower production of HBsAb antibodies and their rapid decrease to non-protective levels [27]. The recommended HBsAb titre following vaccination should preferably be > 100 IU/mL [30] and a booster dose should be administered every year [31]. As reported, a HBsAb level > 10 IU/mL does not always

**Table 5. Predictors of occult hepatitis B virus infection.**

| Variable | OR [95%CI] p-value |
|---|---|
| **Sex** | |
| Female | **Ref** |
| Male | 2.23 [0.41–12.13] 0.352 |
| **Age group** | |
| 21–29 | **Ref** |
| 30–39 | 1 |
| 40–49 | 0.64 [0.04–10.9] 0.760 |
| 50–59 | 1.20 [0.07–20.8] 0.900 |
| 60+ | 4.80 [0.48–47.7] 0.181 |
| **Duration on haemodialysis** | |
| <1 | 2.28 [0.47–11.01] 0.303 |
| 1+ | **Ref** |
| **Cause of CKD** | |
| G.N | 1.77 [0.31–9.98] 0.515 |
| Hypertension + Other | **Ref** |
| **Blood transfusion** | |
| No | |
| Yes | 0.76 [0.08–6.97] 0.808 |
| **Vaccination** | |
| No | 3.18 [0.65–15.54] 0.154 |
| Yes | **Ref** |
| **HBcAb** | |
| Non-reactive | **Ref** |
| Reactive | 82.99 [8.55–806.1] 0.000 |

The first parameter of each variable (indicated as Ref) was used as a reference for the determination of the Odds Ratio and the p-value. *G. N = Glomerulonephritis*.

indicate protection against HBV infection in dialysis patients, and it was suggested that a titre of at least 50 IU/mL should be a target level [32]. Moreover, it is pragmatic that additional HBV doses be administered, if there is no response to the initial HBV vaccination.

An HBcAb prevalence of 12.5% (12/96) was found in this study. The presence of this antibody was the main variable that predicted occult hepatitis B virus infection. OBI showed no significant association with sex, age, blood transfusion, or duration of haemodialysis. Moreover, OBI was found in two of the HBsAb-positive patients and 6 out of the 12 HBcAb-positive patients. This observation is similar to the findings of a study conducted in Iran that found the presence of HBV DNA in 50% of patients on haemodialysis with isolated HBcAb [16]. The high prevalence of OBI among HBcAb-positive patients in our study is also consistent with what was reported in Egypt where a higher frequency of OBI was observed in seropositive subjects [33]. Although the requirement for HBV vaccination in HBV DNA-negative and HBcAb-positive individuals is still controversial, a recent study from China showed that HBcAb-positive subjects maintained titre values of HBsAb similar to those of a control group 8 years after vaccination [34] suggesting that these individuals should also be vaccinated against HBV.

Haemodialysis patients used to be more susceptible to HBV infection because of sharing of dialysis equipment [35]. However, this risk has most likely been eliminated in virtually every country. We found a 7.3% prevalence of OBI among the haemodialysis patients at the Renal Unit of the KBTH. In Iran, a study among patients on maintenance haemodialysis reported an OBI

**Table 6. Characteristics of the HBsAg-negative OBI patients.**

| Sample Code | Sex | Vaccination status | Age (Years) | HBV DNA Viral Load (IU/mL) | HBcAb | HBsAb | HBsAg |
|---|---|---|---|---|---|---|---|
| GT 12 | Male | Yes | 67 | 45 | Positive | Negative | Negative |
| GT 13 | Female | No | 45 | 193 | Positive | Negative | Negative |
| GT 19 | Female | Yes | 60 | 34 | Positive | Positive | Negative |
| GT 39 | Male | Yes | 27 | 120 | Negative | Negative | Negative |
| GT 47 | Male | No | 64 | 95 | Positive | Negative | Negative |
| GT 80 | Male | No | 52 | 83 | Positive | Negative | Negative |
| GT 95 | Male | No | 61 | 121 | Positive | Positive | Negative |
| Mean(SD) | | | 53.7±14.0 | 98.7±53.5 | | | |
| Range | | | | 34–193 | | | |

prevalence of 4.2% [35]. Another study in the same country revealed a 3.1% prevalence of OBI [36]. Ahmadu and colleagues reported prevalence of 7.8% OBI among HD patients in Sudan [37]. A study in Turkey found a 2.7% prevalence of OBI in HD patients [38]. Taken together, these observations illustrate that the prevalence of OBI is relatively low in HD patients.

**Table 7. Prevalence and demographic characteristics of occult hepatitis B virus infection in HBsAg-negative patients.**

| Variable | Occult Hepatitis B status N (%) | | Total | $\chi^2$(p-value) |
|---|---|---|---|---|
| | No = 89 (92.7) | Yes = 7 (7.3) | 96 | |
| **Sex** | | | | 0.91(0.341) |
| Female | 42(95.5) | 2(4.5) | 44 | |
| Male | 47(90.4) | 5(9.6) | 52 | |
| **Age group [Mean(SD) = 45.2(13.9)]** | | | | 7.12(0.130) |
| 21–29 | 18(94.7) | 1(5.3) | 19 | |
| 30–39 | 13(100) | 0(0.0) | 13 | |
| 40–49 | 28(96.6) | 1(3.4) | 29 | |
| 50–59 | 15 (93.8) | 1(6.3) | 16 | |
| '60+ | 15(78.9) | 4(21.1) | 19 | |
| **Duration on haemodialysis [Mean(SD) = 2.3(2.0)]** | | | | 3.41(0.182) |
| <1 | 22(88.0) | 3(12.0) | 25 | |
| 1–2 | 38(90.5) | 4(9.5) | 42 | |
| 3+ | 29(100) | 0(0.0) | 29 | |
| **Cause of CKD** | | | | 1.34(0.86) |
| Glomerulonephritis | 16(88.9) | 2(11.1) | 18 | |
| Hypertension | 61(92.4) | 5(7.6) | 66 | |
| Diabetes | 1 (100) | 0 (0.0) | 1 | |
| Systemic Lupus Erythematosus | 3 (100) | 0 (0.0) | 3 | |
| Other | 8(100) | 0(0) | 8 | |
| **Blood transfusion** | | | | 0.06(0.807) |
| No | 10(90.9) | 1(9.1) | 11 | |
| Yes | 79(92.9) | 6(7.1) | 85 | |
| **Vaccination** | | | | 4.56(0.033) |
| No | 19(82.6) | 4(17.4) | 23 | |
| Yes | 70(95.9) | 3(4.1) | 73 | |
| **HBcAb** | | | | 37.00(0.000) |
| Non-reactive | 83(98.8) | 1(1.2) | 84 | |
| Reactive | 6(50.0) | 6(50.0) | 12 | |

## Conclusions

The prevalence of overt HBV infection among dialysis patients is lower than that among the general Ghanaian population. The HBV occult infection rates in the KBTH Renal Unit are also similar to those reported by studies from diverse geographical locations. The HBcAb-positive parameter, which is indicative of previous HBV infection, is the only clinical laboratory parameter that reliably predicts the presence of occult infections. Immunity against HBV infection is low among the KBTH dialysis patients, with only half of those previously vaccinated being HBsAb-positive and therefore protected against HBV infection. It is therefore, urgent to institute measures in HD Units in Ghana to control and/or prevent HBV infection and transmission. Regular checking of HBsAb titres to determine immunoprotection and re-vaccination of patients if necessary and frequent determination of HBV DNA status will go a long way to reduce possible reactivation of HBV infection. The data reported in our study is limited by the relatively small sample size, therefore further studies with large number of CKD patients on HD is needed to ascertain and implement the findings in this study.

## Supporting information

**S1 File.**
(XLSX)

## Acknowledgments

We thank Mr. John Tetteh for his assistance in the statistical analysis of the data. We also thank the chronic kidney disease patients on haemodialysis who consented to take part in the study and all staff of the Renal Unit at the Korle Bu Teaching Hospital.

## Author Contributions

**Conceptualization:** Bartholomew Dzudzor, Kenneth Tachi.

**Data curation:** Bartholomew Dzudzor, Kingsley Kwame Nsowah.

**Formal analysis:** Vincent Amarh, Kenneth Tachi.

**Funding acquisition:** Bartholomew Dzudzor, Kenneth Tachi.

**Methodology:** Kingsley Kwame Nsowah, Seth Agyemang.

**Project administration:** Bartholomew Dzudzor.

**Supervision:** Bartholomew Dzudzor, Kenneth Tachi.

**Validation:** Bartholomew Dzudzor, Sandro Vento, Vincent Boima.

**Writing – original draft:** Bartholomew Dzudzor.

**Writing – review & editing:** Bartholomew Dzudzor, Kingsley Kwame Nsowah, Seth Agyemang, Sandro Vento, Vincent Amarh, Vincent Boima, Kenneth Tachi.

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
