## [Decision Letter · Decision Letter 0]

9 Oct 2023

PONE-D-23-25809Overt and occult hepatitis B virus infection detected among chronic kidney disease patients on haemodialysis at a Tertiary Hospital in GhanaPLOS ONE

Dear Dr. Dzudzor,

Thank you for submitting your manuscript to PLOS ONE. After careful consideration, we feel that it has merit but does not fully meet PLOS ONE’s publication criteria as it currently stands. Therefore, we invite you to submit a revised version of the manuscript that addresses the points raised during the review process.

We look forward to receiving your revised manuscript.

Kind regards,

Po-Yao Hsu

Academic Editor

PLOS ONE

Journal Requirements:

A clean copy of the edited manuscript (uploaded as the new *manuscript* file)”"

 "-KT and BD

-Partially funded by: University of Ghana Research Fund (UGRF/10/ILG-081/2016-2017) awarded to KT (main investigator) and BD (co-investigator).

-No. The funders had no role in study design, data collection and analysis, decision to publish, or preparation of the manuscript."

5. We note you have included a table to which you do not refer in the text of your manuscript. Please ensure that you refer to Table 5 in your text; if accepted, production will need this reference to link the reader to the Table.

**Additional Editor Comments:**

The authors demonstrated the prevalence of overt and occult HBV infection among patients on hemodialysis at a Tertiary Hospital in Ghana. I have some comments as follows:

1. Table 1 should present the clinical features of all patients (n=104).

2. The titles of other tables need to show which category (HBsAg-negative or HBsAg-positive) the patients belong to.

3. The patient number of this study was relatively small. It should be described as a limitation.

Reviewers' comments:

Reviewer's Responses to Questions

**Comments to the Author**

1. Is the manuscript technically sound, and do the data support the conclusions?

Reviewer #1: Yes

Reviewer #2: Yes

2. Has the statistical analysis been performed appropriately and rigorously? 

Reviewer #1: Yes

Reviewer #2: Yes

3. Have the authors made all data underlying the findings in their manuscript fully available?

Reviewer #1: Yes

Reviewer #2: Yes

4. Is the manuscript presented in an intelligible fashion and written in standard English?

Reviewer #1: Yes

Reviewer #2: No

5. Review Comments to the Author

Reviewer #1: Thank you for the opportunity to review this manuscript. Recipients of regular hemodialysis are at increased risk of viral hepatitis infection on account of their immunosuppressed state and their frequent exposure to invasive clinical procedures. Accordingly, the authors of this manuscript sought to evaluate the prevalence of overt and occult hepatitis B virus infection among chronic kidney disease patients on hemodialysis at a hospital in Ghana. The findings of this study contribute further evidence for informing reforms to the national guidelines for the management, prevention, and control of viral hepatitis among at-risk groups including hemodialysis patients. While the study has merit, I note several omissions in the reporting of the methods and results. The following comments need addressing:

1. Given the disproportionate risk of HBV infection among hemodialysis patients, the authors should address the availability of national policies or guidelines for prevention and control of hepatitis B among general and at-risk populations including hemodialysis patients. Further to this, the authors do not address the burden of chronic hepatitis B or hepatitis C or HBV-HCV co-infection in Ghana. Neither do they address the national hepatitis B vaccination program which are all relevant in providing context for this study (Introduction section).

2. For all assays used, please provide the sensitivity / assay cut-off values / lower limit of detection where relevant (Method section).

3. The sampling and the data are old (between September 17, 2017 and December 21, 2017). The current information on prevalence of HBV infection has been surely changed (Method section).

4. This study is not controlled. The findings from the hemodialysis patient group is not compared with that from the control group. None of the tables provided in this manuscript provide data on the control population. Please add the demographic, clinical and study findings of the control group, this significantly impacts on the interpretation of the overall findings (Result section).

Reviewer #2: Authors should be able to distinguish between false OBI, patients who might be in the reactivation phase of occult hepatitis,

and those who are truly positive for HBsAg but could not be detected by the ELISA system.

Indeed, during the reactivation phase of OBI, HBsAg is sometimes negative in the first weeks.

6. PLOS authors have the option to publish the peer review history of their article (what does this mean?). If published, this will include your full peer review and any attached files.

Reviewer #1: **Yes: **Fatemeh Farshadpour

Reviewer #2: No

---

## [Author Response · Author response to Decision Letter 0]

12 Jan 2024

Response to Journal’s requirements:

Response: The revised manuscript has been formatted in accordance with PLOS ONE’s style requirements.

2. We suggest you thoroughly copyedit your manuscript for language usage, spelling, and grammar. If you do not know anyone who can help you do this, you may wish to consider employing a professional scientific editing service. Upon resubmission, please provide the following:

Response: All the authors have thoroughly reviewed and edited the revised manuscript for language usage, spelling and grammar.

Response: A copy of the revised manuscript showing the changes (using track changes) has been uploaded as a *supporting information* file

A clean copy of the edited manuscript (uploaded as the new *manuscript* file)”"

Response: A clean copy of the revised manuscript has been uploaded as the new *manuscript* file.

 "-KT and BD

-Partially funded by: University of Ghana Research Fund (UGRF/10/ILG-081/2016-2017) awarded to KT (main investigator) and BD (co-investigator).

-No. The funders had no role in study design, data collection and analysis, decision to publish, or preparation of the manuscript."

Please provide an amended statement that declares *all* the funding or sources of support (whether external or internal to your organization) received during this study, as detailed online in our guide for authors at http://journals.plos.org/plosone/s/submit-now. 

Response: The declaration statement for the funding or sources of support received during this study has been amended, as suggested by the Editor. 

Please also include the statement “There was no additional external funding received for this study.” in your updated Funding Statement. 

Response: The statement ‘There was no additional external funding received for this study’ has been included in the updated Funding Statement in the revised manuscript.

Response: The amended Funding Statement has been included in the cover letter. 

4. In your Data Availability statement, you have not specified where the minimal data set underlying the results described in your manuscript can be found. 

Response: The study’s minimal underlying dataset has been uploaded as a Supporting Information file and the URL link insert in the cover letter as requested. All patient information has been fully anonymized.

Response: The authors are grateful for this kind gesture.

5. We note you have included a table to which you do not refer in the text of your manuscript. Please ensure that you refer to Table 5 in your text; if accepted, production will need this reference to link the reader to the Table.

Response: Table 5 has been referenced in the revised manuscript (Line 229- 231). 

Response: The authors have reviewed the reference list and can confirm that it is complete and correct. Moreover, none of the articles cited in the manuscript has been retracted. Furthermore, authors have also added three references to support the sensitivity and specificity of the kits used in the analysis (ref number 18 through to 20).

Response to additional comments from the Editor:

The authors demonstrated the prevalence of overt and occult HBV infection among patients on hemodialysis at a Tertiary Hospital in Ghana. I have some comments as follows:

1. Table 1 should present the clinical features of all patients (n=104).

Response: The initial 104 participants were tested for the presence of HBsAg. Age, gender, HBsAg-viral load was determined for all using the Elecsys from Roche Diagnostics. The structured questionnaire captured the demographic information and other clinical features, i.e. vaccination and blood transfusion status of all the patients. 

Tables 1 has been edited to provide clarity on the above explanations. The Methods and Results sections have also been edited accordingly (Lines 110 – 125 and 180 – 185).

2. The titles of other tables need to show which category (HBsAg-negative or HBsAg-positive) the patients belong to.

Response: The titles of Tables 2 – 7 have been edited to indicate the category of the study patients with respect to their HBsAg status.

3. The patient number of this study was relatively small. It should be described as a limitation.

Response: The relatively small sample size has been indicated as a limitation at the conclusion section of the revised manuscript (Lines 334 – 337).

Response to comments from Reviewer #1: 

1. Given the disproportionate risk of HBV infection among hemodialysis patients, the authors should address the availability of national policies or guidelines for prevention and control of hepatitis B among general and at-risk populations including hemodialysis patients. Further to this, the authors do not address the burden of chronic hepatitis B or hepatitis C or HBV-HCV co-infection in Ghana. Neither do they address the national hepatitis B vaccination program which are all relevant in providing context for this study (Introduction section).

Response: Our focus is mainly on HBV infection among hemodialysis patients. We did not look at HCV or HBV-HCV co-infection because of our focus. This suggestion is welcomed and can be considered in our follow-up comprehensive study which will factor in other infectious agents. The need for relevant policies to reduce or prevent HBV infection has been suggested in the conclusion section (lines 330 -334)

2. For all assays used, please provide the sensitivity / assay cut-off values / lower limit of detection where relevant (Method section).

Response: This has been included in the method section (lines 128-129; 136-137, 139-140; and 155-156)

3. The sampling and the data are old (between September 17, 2017 and December 21, 2017). The current information on prevalence of HBV infection has been surely changed (Method section).

Response: The data presented in this manuscript were obtained from samples collected approximately 6 years ago. Even though the authors agree that the current prevalence of HBV infection might have changed, we are convinced that the data in this manuscript is a vital reference for subsequent studies on HBV infection in Ghana. 

4. This study is not controlled. The findings from the hemodialysis patient group is not compared with that from the control group. None of the tables provided in this manuscript provide data on the control population. Please add the demographic, clinical and study findings of the control group, this significantly impacts on the interpretation of the overall findings (Result section).

Response: The authors agree with the reviewer that data for a control (non-haemodialysis) group were not included in the study design. We are also conscious that inclusion of a control group will significantly impact the interpretation of the data. Nonetheless, the relevant statistical analysis were performed for the data illustrated in Tables 4, 5 and 7. Moreover, suitable groups were used as reference for the estimation of odds ratio in Table 5. Hence, the authors are of the opinion that the dataset in the manuscript are in support of the overall objectives of the study. Furthermore, this is a hospital-based cross-sectional study. 

Response to comments from Reviewer #2: 

Authors should be able to distinguish between false OBI, patients who might be in the reactivation phase of occult hepatitis, and those who are truly positive for HBsAg but could not be detected by the ELISA system. Indeed, during the reactivation phase of OBI, HBsAg is sometimes negative in the first weeks.

Response: In true OBI cases, HBV DNA in serum/plasma is intermittent, and when detectable, it is usually lower than 200 IU/mL. In fact, HBV-DNA levels were below 200 IU/ml in all our cases.

“False OBI” should be considered when serum HBV DNA levels are comparable to those usually detected in cases of overt HBV infection. This did not occur in any of our cases.

The diagnosis of HBV reactivation in OBI patients can be made in the case of (a) HBsAg reappearance and/or at least 1 log increase above the LLOD of serum HBV DNA in a subject with previously undetectable HBsAg and serum HBV DNA and (b) at least 1 log increase of serum HBV DNA in a subject with previously detectable HBV DNA [Raimondo , Locarnini S, Pollicino T et al. Taormina Workshop on Occult HBV Infection Faculty Members. Update of the statements on biology and clinical impact of occult hepatitis B virus infection. J Hepatol 2019; 71: 397–408]. Our study was cross-sectional, so we could not evaluate this possibility.

---

## [Decision Letter · Decision Letter 1]

19 Feb 2024

Overt and occult hepatitis B virus infection detected among chronic kidney disease patients on haemodialysis at a Tertiary Hospital in Ghana

PONE-D-23-25809R1

Dear Dr. Dzudzor,

We’re pleased to inform you that your manuscript has been judged scientifically suitable for publication and will be formally accepted for publication once it meets all outstanding technical requirements.

Kind regards,

Po-Yao Hsu

Academic Editor

PLOS ONE

Additional Editor Comments (optional):

All comments have been addressed.

Reviewers' comments:

Reviewer's Responses to Questions

**Comments to the Author**

1. If the authors have adequately addressed your comments raised in a previous round of review and you feel that this manuscript is now acceptable for publication, you may indicate that here to bypass the “Comments to the Author” section, enter your conflict of interest statement in the “Confidential to Editor” section, and submit your "Accept" recommendation.

Reviewer #1: All comments have been addressed

2. Is the manuscript technically sound, and do the data support the conclusions?

Reviewer #1: Yes

3. Has the statistical analysis been performed appropriately and rigorously? 

Reviewer #1: I Don't Know

4. Have the authors made all data underlying the findings in their manuscript fully available?

Reviewer #1: Yes

5. Is the manuscript presented in an intelligible fashion and written in standard English?

Reviewer #1: Yes

6. Review Comments to the Author

Reviewer #1: (No Response)

7. PLOS authors have the option to publish the peer review history of their article (what does this mean?). If published, this will include your full peer review and any attached files.

Reviewer #1: **Yes: **Fatemeh Farshadpour

---

## [Editor Report · Acceptance letter]

23 Feb 2024

PONE-D-23-25809R1 

PLOS ONE

Dear Dr. Dzudzor, 

I'm pleased to inform you that your manuscript has been deemed suitable for publication in PLOS ONE. Congratulations! Your manuscript is now being handed over to our production team.

Kind regards, 

on behalf of

Dr. Po-Yao Hsu 

Academic Editor

PLOS ONE